# *ZSCAN4* Regulates Zygotic Genome Activation and Telomere Elongation in Porcine Parthenogenetic Embryos

**DOI:** 10.3390/ijms241512121

**Published:** 2023-07-28

**Authors:** Xiao-Han Li, Ming-Hong Sun, Wen-Jie Jiang, Dongjie Zhou, Song-Hee Lee, Geun Heo, Zhi Chen, Xiang-Shun Cui

**Affiliations:** 1Department of Animal Science, Chungbuk National University, Cheongju 28644, Republic of Korea; 2College of Animal Science and Technology, Yangzhou University, Yangzhou 225009, China

**Keywords:** *ZSCAN4*, *DNMT1*, DNA methylation, telomere length, ZGA, porcine embryos

## Abstract

Zinc finger and SCAN domain-containing 4 (*ZSCAN4*), a DNA-binding protein, maintains telomere length and plays a key role in critical aspects of mouse embryonic stem cells, including maintaining genomic stability and defying cellular senescence. However, the effect of *ZSCAN4* in porcine parthenogenetic embryos remains unclear. To investigate the function of *ZSCAN4* and the underlying mechanism in porcine embryo development, *ZSCAN4* was knocked down via dsRNA injection in the one-cell stage. *ZSCAN4* was highly expressed in the four- and five- to eight-cell stages in porcine embryos. The percentage of four-cell stage embryos, five- to eight-cell stage embryos, and blastocysts was lower in the *ZSCAN4* knockdown group than in the control group. Notably, depletion of *ZSCAN4* induced the protein expression of *DNMT1* and 5-Methylcytosine (5mC, a methylated form of the DNA base cytosine) in the four-cell stage. The H3K27ac level and ZGA genes expression decreased following *ZSCAN4* knockdown. Furthermore, *ZSCAN4* knockdown led to DNA damage and shortened telomere compared with the control. Additionally, *DNMT1*-dsRNA was injected to reduce DNA hypermethylation in *ZSCAN4* knockdown embryos. *DNMT1* knockdown rescued telomere shortening and developmental defects caused by *ZSCAN4* knockdown. In conclusion, *ZSCAN4* is involved in the regulation of transcriptional activity and is essential for maintaining telomere length by regulating *DNMT1* expression in porcine ZGA.

## 1. Introduction

The process by which a mammalian oocyte develops into a blastocyst after fertilization is known as preimplantation embryonic development [1]. Zygotic genome activation (ZGA) is a critical stage in the development of preimplantation mammalian embryos in the two-cell stage in mice, four-cell stage in pigs [2], and five- to eight-cell stages in cattle and humans [3,4]. During the ZGA period, maternal mRNAs are degraded, and a large number of zygotic genes are transcribed [5]. Zygotic genome transcription failure results in embryonic development arrest [6,7].

The zinc finger and SCAN domain-containing 4 (*ZSCAN4*) gene was originally discovered in a small number of embryonic stem cells and in two-cell stage mouse embryos [8]. In mouse embryonic stem cells (mESCs), *ZSCAN4* expression is transient; it is expressed in only 1% to 5% of ES cells at any given time point [9]. The transient expression pattern of *ZSCAN4* is accompanied by key cellular events, including heterochromatin transcription bursts, DNA methylation, and histone acetylation [10]. During *ZSCAN4* transcription, other ZGA-specific genes are also expressed at high levels [11]. For example, *Eif1a*, *Tcstv1/3*, and *Tbx3* are more highly expressed in *Zscan4*^+^ ES cells than in *Zscan4*^−^ ES cells [11,12,13]. *ZSCAN4* is essential for the development of embryonic stem cells and mouse preimplantation embryos. *ZSCAN4* knockdown (KD) causes most ES cells to stop proliferating and increases the level of apoptosis at passage 7 [9]. Moreover, the KD of *ZSCAN4* using siRNA in mouse embryos delays two- to four-cell embryo development and leads to implantation failure [8]. Bovine embryos failed to develop to the 16-cell stage after the injection of *ZSCAN4* siRNA [14]. *ZSCAN4* has been shown to regulate multiple biological events. It can promote the generation of induced pluripotent stem cells [15]. The *ZSCAN4*–TET2 interaction can regulate the expression of glycolytic enzymes and proteasome subunit-related genes, and thereby enhance proteasome function and regulate metabolic rewiring [16]. In human head and neck squamous cell carcinoma, depletion of *ZSCAN4* downregulates the expression of cancer stem cell (CSC) markers, severely affecting tumorsphere formation and tumor growth [17]. In addition, an important function of *ZSCAN4* is the regulation of telomere elongation and maintenance of genomic stability [8,9,18].

Mammalian telomeres consists of hundreds to thousands of short, repetitive nucleotide sequences and telomere-associated binding proteins [19,20]. An appropriate telomere length is required for mammalian cell proliferation, and short telomeres cause DNA damage, chromosome fusion, cellular senescence, and apoptosis [21]. Telomere length is influenced by changes in telomerase activity and alternative lengthening of telomere (ALT) pathway [22,23]. Telomerase activity is low or absent in mature mammalian oocytes and early embryos compared with that in stem cells and male germlines with high telomerase activity, and is not elevated again until the blastocyst stage [24]. Telomere length maintenance during oocyte maturation after fertilization and early embryonic development is regulated by the ALT pathway, and in the blastocyst stage, it is regulated by telomerase [25]. In recent years, the function of *ZSCAN4* in telomere length maintenance has been reported. In human ALT cancer cells, *ZSCAN4* is highly expressed, and its expression is critical for extending these short telomeres [22]. In mouse embryonic stem cells, *ZSCAN4* induces DNA demethylation to maintains telomere length [26]. However, the expression pattern and effect of *ZSCAN4* during porcine parthenogenetic embryo development have rarely been studied.

Previous reports have shown that parthenogenetic porcine oocytes that had been matured in vitro can develop to the blastocyst, stage and the characteristics of the development of porcine parthenogenetic embryos to the blastocyst stage resembled those of in vitro-fertilized embryos [27]. It is difficult to obtain porcine embryos of homogeneous quality due to the relatively high incidence of polyspermy that occurs during in vitro fertilization. Therefore, porcine parthenogenetic embryos have frequently been used to study early development [28]. In this study, we hypothesized that *ZSCAN4* is crucial for the ZGA stage and telomere elongation in porcine parthenogenetic embryos. To evaluate this hypothesis, *ZSCAN4* was knocked down using double-stranded RNA (dsRNA) at the zygote stage and cultured until the four-cell stage for further analysis.

## 2. Results

### 2.1. Expression and Localization of ZSCAN4 in Porcine Embryos

First, we examined the mRNA expression level of *ZSCAN4* during porcine embryonic development. qRT-PCR was used to detect *ZSCAN4* expressed in different stages of embryonic development. We found that *ZSCAN4* was highly expressed at the four- and eight-cell stages, and its expression significantly decreased in the morula and blastocyst stages (Figure 1A). This expression pattern indicates that *ZSCAN4* might play a role in porcine ZGA. Subsequently, the subcellular localization of *ZSCAN4* in porcine embryos was examined using immunofluorescence staining. *ZSCAN4* had no obvious signal in two-cell embryos and blastocysts, and *ZSCAN4* was enriched in the nucleus in the four- and eight-cell embryos (Figure 1B).

### 2.2. Effects of ZSCAN4 KD on Porcine Embryonic Development 

We used dsRNA to KD *ZSCAN4* and explored the prospective functions of *ZSCAN4* during porcine embryonic development. The blastocyst rate of embryos was not significantly different between the GFP-dsRNA group and the NF-water group (GTP-dsRNA vs. NF-water, 40.68 ± 4.575, *n* = 251 vs. 40.15 ± 3.814, *n* = 252; *p* > 0.05). Then, we used NF-water injection as the control group. The efficiency of *ZSCAN4* KD was evaluated in the four-cell stage using qRT-PCR assay, and the results showed that the injection of *ZSCAN4* dsRNA effectively reduced the *ZSCAN4* mRNA level (control vs. *ZSCAN4* KD, 1.0 vs. 0.61 ± 0.025; *p* < 0.001) (Figure 2B). The knockdown of *ZSCAN4* was verified using Western blotting in the four-cell stage. We found that the expression of *ZSCAN4* was also reduced after *ZSCAN4* KD (control vs. *ZSCAN4* KD, 1.0 vs. 0.61 ± 0.067; *p* < 0.01) (Figure 2C). Next, we collected two-cell, four-cell, eight-cell, and blastocyst-stage embryos to calculate the cleavage rate after *ZSCAN4* KD. We found that the cleavage rate of one- to two-cell embryos did not differ between the control and *ZSCAN4* KD groups. However, the formation of four-cell embryos, eight-cell embryos, and blastocysts was decreased after *ZSCAN4* KD (two-cell stage, control: 75.8%  ±  3.95%, *n* = 270, vs. *ZSCAN4* KD: 71.0%  ±  5.76%, *n* = 225, *p* > 0.05; four-cell stage, control: 63.6%  ±  1.82%, *n* = 405, vs. *ZSCAN4* KD: 46.7%  ±  2.42%, *n* = 315, *p* < 0.01; eight-cell stage, control: 50.3%  ±  4.87%, *n* = 182 vs. *ZSCAN4* KD: 38.5%  ±  5.29%, *n* = 185, *p* < 0.05; blastocyst stage, control: 44.6% ± 3.67%, *n* = 102 vs. *ZSCAN4* KD: 28.9% ± 2.33%, *n* = 117, *p* < 0.001) (Figure 2D,E). Moreover, we determined the total cell number and diameter in the blastocyst stage after *ZSCAN4* KD. The total cell number and diameter were reduced in the *ZSCAN4* KD group compared with those in the control group (cell number, control: 36.2 ± 1.73, *n* = 9 vs. *ZSCAN4* KD: 18.9 ± 1.88, *n* = 9; *p* < 0.001; diameter, control: 205.4 ± 13.8, *n* = 14, vs. *ZSCAN4* KD: 165.7 ± 5.57 μm, *n* = 19; *p* < 0.05) (Figure 2F). 

### 2.3. Effect of ZSCAN4 KD on Histone Modifications and ZGA 

To analyze the effect of *ZSCAN4* on histone modifications related to transcriptional activity, we examined changes in histone modifications, mainly histone methylation and acetylation, after *ZSCAN4* KD. Western blotting was used to determine the levels of H3K9me3 and H3K27ac after *ZSCAN4* KD. The Western blot analysis showed that the expression of H3K9me3 was considerably higher in *ZSCAN4* KD embryos than in the control group (control: *ZSCAN4* KD, 1.0 vs. 1.20 ± 0.0362; *p* < 0.05) (Figure 3A). We also examined the H3K9me3 level using immunofluorescence staining for further verification (control: *ZSCAN4* KD, 41.00 ± 3.17, *n* = 96 vs. 51.8 ± 3.32, *n* = 114; *p* < 0.01) (Figure 3C). Next, we found that *ZSCAN4* KD decreased the level of H3K27ac in the Western blot analysis (control: *ZSCAN4* KD, 1.0 vs. 0.673 ± 0.0815; *p* < 0.05) (Figure 3B). The fluorescence signals decreased in the *ZSCAN4* KD embryos compared with those in the control group. The fluorescence intensity analysis also confirmed the findings (control: *ZSCAN4* KD, 44.6 ± 4.84, *n* = 71 vs. 37.8 ± 4.17, *n* = 67; *p* < 0.01) (Figure 3D), which was consistent with the Western blotting results. Next, to explore the effect of *ZSCAN4* on the ZGA, we detected changes in ZGA genes using RT-qPCR. ZGA genes *eIF1a*, *Tbx3*, *Tcstv3*, *Rif1*, *Wee1,* and *Dppa2* in the *ZSCAN4* KD group were significantly lower than those in the control group (*eIF1a*, control: 1.0 vs. *ZSCAN4* KD: 0.470  ±  0.103, *p* < 0.01; *Tbx3*, control: 1.0 vs. *ZSCAN4* KD: 0.367  ±  0.0820, *p* < 0.05; *Tcstv3*, control: 1.0 vs. *ZSCAN4* KD: 0.323  ±  0.0451, *p* < 0.001; *Dppa2*, control: 1.0 vs. *ZSCAN4* KD: 0.571  ±  0.0601, *p* < 0.001; *Rif1*, control: 1.0 vs. *ZSCAN4* KD: 0.281  ±  0.0233, *p* < 0.01; *Wee1*, control: 1.0 vs. *ZSCAN4* KD: 0.539  ±  0.0861, *p* < 0.01) (Figure 3E). The results partially demonstrated that *ZSCAN4* regulates histone methylation and acetylation, which are critical for the activation of ZGA gene expression in the ZGA stage in porcine embryos.

### 2.4. ZSCAN4 KD Induced Global DNA Methylation

To investigate the effect of *ZSCAN4* KD in early porcine embryos, qRT-PCR was used to determine the mRNA level of *DNMT1*. After *ZSCAN4* KD, *DNMT1* mRNA expression increased (control: *ZSCAN4* KD, 1.0 vs. 1.85± 0.117; *p* < 0.001) (Figure 4A). We also determined the protein levels of *DNMT1* and TET2 in the four-cell stage using Western blotting. *DNMT1* expression increased and TET2 expression decreased after *ZSCAN4* KD (control: *ZSCAN4* KD, *DNMT1*:1.0 vs. 1.30 ± 0.122; *p* < 0.05; TET2:1.0 vs. 0.485 ± 0.0791; *p* < 0.01) (Figure 4B,C). Moreover, the results of fluorescent staining of *DNMT1* proved that the KD of *ZSCAN4* led to increased expression of *DNMT1* (control: *ZSCAN4* KD, 23.5 ± 0.963, *n* = 40, vs. 38.30 ± 1.45, *n* = 32; *p* < 0.001) (Figure 4D). Next, we determined the 5 mc level in the four-cell stage using immunofluorescence staining. We found that four-cell embryos in the *ZSCAN4* KD group had higher fluorescent signals than the control group (control: *ZSCAN4* KD, 19.2 ± 1.11, *n* = 45, vs. 31.0 ± 1.74, *n* = 43; *p* < 0.001) (Figure 4E). These results demonstrate that *ZSCAN4* might regulate DNA methylation by affecting the expression of *DNMT1* and TET2.

### 2.5. ZSCAN4 Regulated DNMT1 Expression to Stabilize Telomere Length

To analyze the potential mechanism of *ZSCAN4* in telomere elongation in porcine embryos, we knocked down *DNMT1* based on *ZSCAN4* KD. First, we detected the expression of *DNMT1* using Western blotting and found that *DNMT1* KD effectively reduced the high level of *DNMT1* induced by *ZSCAN4* depletion (control, 1.0; *ZSCAN4* KD, 1.15 ± 0.0343, *p* < 0.05; *ZSCAN4* KD + *DNMT1* KD, 0.684 ± 0.140, *p* < 0.05) (Figure 5A). The results of fluorescent staining of *DNMT1* confirmed our findings (control, 31.0 ± 2.64, *n* = 130; *ZSCAN4* KD, 43.3 ± 2.52, *n* = 133, *p* < 0.05; *ZSCAN4* KD + *DNMT1* KD, 32.2 ± 3.31, *n* = 129, *p* < 0.05) (Figure 5B). Next, we detected the expression of 5mc using staining. The fluorescence signal of 5mc in the *ZSCAN4* KD group was higher than that in the control group. However, the signal in the double KD group was significantly lower than that in the *ZSCAN4* KD group. Our results showed that *DNMT1* KD effectively reduces the high level of 5mc induced by *ZSCAN4* depletion (control, 14.9 ± 0.945, *n* = 75; *ZSCAN4* KD, 23.6 ± 0.706, *n* = 84, *p* < 0.05; *ZSCAN4* KD + *DNMT1* KD, 18.2 ± 1.44, *n* = 61, *p* < 0.05) (Figure 5C). Moreover, we found that the ratio of four-cell embryos and blastocysts in the double KD group were higher than that in the *ZSCAN4* KD group. It shows that *DNMT1* KD effectively rescues embryonic development arrest caused by *ZSCAN4* depletion (four-cell: control, 57.8% ± 2.77%, *n* = 640; *ZSCAN4* KD, 42.9% ± 2.81%, *n* = 726, *p* < 0.001; *ZSCAN4* KD + *DNMT1* KD, 54.3% ± 3.26%, *n* = 729, *p* < 0.001; blastocyst: control, 25.8% ± 1.82%, *n* = 370; *ZSCAN4* KD, 18.4% ± 2.53%, *n* = 372, *p* < 0.001; *ZSCAN4* KD + *DNMT1* KD, 21.7% ± 2.47%, *n* = 359, *p* < 0.001) (Figure 5D). To explore the effects of *ZSCAN4* and *DNMT1* on telomere length, we examined the relative length of telomeres in four-cell porcine embryos after *ZSCAN4* KD and *DNMT1* KD using qPCR. We found that the relative telomere length in the *ZSCAN4* KD group was shorter than that in the control group. However, the relative telomere length in the double KD group was longer than that in the *ZSCAN4* KD group (control, 1.0; *ZSCAN4* KD, 0.555 ± 0.0828, *p* < 0.01; *ZSCAN4* KD + *DNMT1* KD, 0.836 ± 0.0428, *p* < 0.05) (Figure 5E). Our results showed that *ZSCAN4* KD shortened telomeres, and *DNMT1* KD effectively rescued the telomere shortening induced by *ZSCAN4* depletion. Promyelocytic leukemia protein (PML) is required for the ALT pathway [29]. To further verify the changes in telomere length, PML was quantified as an ALT biomarker using Western blotting. We found that the expression of PML was decreased after *ZSCAN4* KD, and *DNMT1* KD effectively elevated *ZSCAN4* depletion-induced low levels of PML (control, 1.0; *ZSCAN4* KD, 0.584 ± 0.0611, *p* < 0.01; *ZSCAN4* KD + *DNMT1* KD, 1.11 ± 0.231, *p* < 0.05) (Figure 5F). The results of fluorescent staining of PML also confirmed our findings (control, 42.8 ± 5.57, *n* = 138; *ZSCAN4* KD, 36.0 ± 4.85, *n* = 104, *p* < 0.01; *ZSCAN4* KD + *DNMT1* KD, 38.8 ± 5.22, *n* = 104, *p* < 0.05) (Figure 5G). Our results suggest that *ZSCAN4* maintains telomere length by repressing *DNMT1* expression, thereby promoting the development of porcine embryos.

### 2.6. ZSCAN4 KD Induced DNA Damage and Apoptosis in Porcine Embryos

To analyze the effect of *ZSCAN4* on genomic stability in the ZGA stage in porcine embryos, DNA damage was detected using pH2A.X antibody. Western blotting was used to determine the level of pH2A.X after *ZSCAN4* KD. The expression of pH2A.X was higher in *ZSCAN4* KD embryos than in the control group (control: *ZSCAN4* KD, 1.0 vs. 1.57 ± 0.0688; *p* < 0.05) (Figure 6A). In the staining analysis, DNA damage in *ZSCAN4* KD was higher than that in the control group (control vs. *ZSCAN4* KD, 24.5 ± 2.14, *n* = 126 vs. 26.5 ± 2.26, *n* = 101; *p* < 0.05) (Figure 6B). Apoptosis is a prominent mode of cell death following the induction of DNA damage [30]. To verify the occurrence of apoptosis, qRT-PCR was used to determine the mRNA level of apoptosis-related genes (*Bcl2*, *Bax*, and *Caspase3*). Changes in the expression levels of the apoptosis-related genes indicated apoptosis in *ZSCAN4* KD embryos (*Bcl2*, control: 1.0 vs. *ZSCAN4* KD: 0.547 ± 0.0790, *p* < 0.01; *Bax*, control: 1.0 vs. *ZSCAN4* KD: 1.30 ± 0.0608, *p* < 0.05; *Caspase3*, control: 1.0 vs. *ZSCAN4* KD: 1.35 ± 0.0862, *p* < 0.01) (Figure 6C). Furthermore, active *Caspase3* was quantified as an apoptotic biomarker. The total apoptosis was increased in the *ZSCAN4* KD group compared with that in the control group (control vs. *ZSCAN4* KD, 32.3 ± 0.737, *n* = 58, vs. 41.5 ± 1.58, *n* = 51; *p* < 0.001) (Figure 6D). The p53 tumor suppressor protein response to DNA damage or checkpoint failure triggers a series of antiproliferative responses. One of the important functions of p53 is to induce apoptosis [30,31]. We determined the level of p53 in the four-cell stage using immunofluorescence staining. We found that four-cell embryos in the *ZSCAN4* KD group had higher fluorescent signals than the control group (control: *ZSCAN4* KD, 30.3 ± 1.65, *n* = 73, vs. 38.1 ± 1.59, *n* = 58; *p* < 0.01) (Figure 6E). 

## 3. Materials and Method

All chemicals were purchased from Sigma (Sigma-Aldrich, St. Louis, MO, USA) unless otherwise indicated. 

### 3.1. ZSCAN4 dsRNA and DNMT1 dsRNA Preparation

*ZSCAN4* and *DNMT1* were amplified using polymerase chain reaction (PCR) with a pair of primers including the T7 sequence (Table 1). Double-stranded RNA was synthesized using the purified PCR products and MEGAscript T7 Kit (AM1333; Ambion, Huntingdon, UK) according to the previous instructions [32]. DNase I and RNase A were added to remove the DNA templates after in vitro transcription. Next, dsRNA was purified using phenol–chloroform and isopropyl alcohol. The purified dsRNA was dissolved in RNase-free water and stored at −80 °C until use.

### 3.2. Oocyte Harvest and In Vitro Maturation

Porcine ovaries were obtained from a local slaughterhouse and placed in 38 °C saline supplemented with 50 mg/mL streptomycin sulfate and 75 mg/mL penicillin G. Follicular fluid was collected using a syringe with a 6-gauge needle, and cumulus–oocyte complexes with at least three layers of dense cumulus cells and a uniformly granular ooplasm were collected under a microscope. A total of 500 μL of in vitro maturation medium [TCM-199 (Invitrogen, Carlsbad, CA, USA) supplemented with 0.57 mM L-cysteine (Sigma), 10 IU/mL follicle-stimulating hormone (Sigma), 10 ng/mL epidermal growth factor (Sigma), 10 IU/mL luteinizing hormone (Sigma), 0.1 mg/mL sodium pyruvate (Sigma), and 10% (*v*/*v*) porcine follicular fluid (Sigma)] per well was added to a 4-well plate. After rinsing in TL-HEPES, approximately 80 COCs were transferred to each well. The 4-well plate (30004, SPL Life Sciences, Seoul, Republic of Korea) was placed in an incubator maintained at 38.5 °C in an atmosphere of 5% CO_2_ and 100% humidity after covering it with mineral oil (370 μL/well).

### 3.3. Parthenogenetic Activation, In Vitro Culture, and dsRNA Injection

After 48 h of culture, 1 mg/mL hyaluronidase was used to remove the cumulus cells. Fresh MII oocytes were parthenogenetically activated using two direct-current pulses of 120 V for 60 µs in 297 mmol/L mannitol (pH 7.2) containing 0.01% polyvinyl alcohol (PVA, *w*/*v*), 0.5 mmol/L HEPES, 0.05 mmol/L MgSO_4_, and 0.1 mmol/L CaCl_2_. To suppress extrusion of the pseudo-second polar body, these oocytes were cultured in bicarbonate-buffered porcine zygote medium 5 (PZM-5) containing 7.5 µg/mL cytochalasin B and 4 mg/mL bovine serum albumin (BSA) for 3 h. In the knockdown group, each activated oocyte was microinjected with 5–10 pl dsRNA using an Eppendorf Femto-Jet (Eppendorf, Hamburg, Germany) under a Nikon Diaphot Eclipse TE300 inverted microscope (Nikon, Tokyo, Japan) after thorough washing. The dsRNA concentration used for microinjection was 1200 ng/μL. Nucleus-free water was injected to the oocytes in the control group. The oocytes were then transferred to in vitro culture medium (bicarbonate-buffered PZM-5 supplemented with 4 mg/mL BSA) and incubated for 2 days at 38.5 °C under 5% CO_2_.

### 3.4. Quantitative Reverse Transcription PCR (qRT-PCR)

Quantitative reverse transcription PCR was used to evaluate gene expression in the embryos. According to the manufacturer’s instructions, mRNA was extracted from 40 embryos in each group using the Dynabeads mRNA Direct Kit (61012; Thermo Fisher Scientific, CA, USA). cDNA was reverse-transcribed using the First-Strand Synthesis Kit (6210; LeGene, San Diego, CA, USA). Next, qRT-PCR was performed using the WizPure™ qPCR Master Mix (Cat # W1731-8; Wizbiosolution, Seongnam, Republic of Korea). The qRT-PCR protocol was as follows: 95 °C for 3 min, followed by 40 cycles at 95 °C for 15 s, 60 °C for 25 s, and 72 °C for 10 s, with a final extension at 72 °C for 5 min. The target genes were *ZSCAN4*, *DNMT1*, *EIF1A*, *CASPASE 3*, *BCL2*, *BAX*, *TCSTV3*, *DPPA2*, *TBX3*, *RIF1*, and *WEE1*. *18S* rRNA was used as the reference gene. The mRNA expression was analyzed using the 2^−ΔΔCT^ method. For telomere measurement, approximately 100 embryos in the four-cell stage per group were collected, and genomic DNA was extracted using the DNeasy Blood & Tissue Kit (Qiagen, Valencia, CA, USA). Average telomere length was measured from total genomic DNA using qPCR. The qPCR protocol was as follows: 95 °C for 10 min, followed by 40 cycles at 95 °C for 15 s, 60 °C for 60 s, and 72 °C for 15 s, with a final extension at 72 °C for 5 min. The primers used are listed in Table 1. 

### 3.5. Immunofluorescence and Confocal Microscopy

Four-cell embryos were fixed in 3.7% formaldehyde for 1 h at 25 °C. The embryos were then permeabilized in 0.5% Triton-X100 for 1 h at 25 °C. Next, the embryos were transferred to 3% bovine serum albumin (BSA) for 1 h after three washes in PBS/PVA. For 5mc staining, the embryos were fixed in ice-cold 70% ethanol for 5 min. The fixed sample was incubated in 1.5 M HCL for 1 h after rinsing three times in 1× PBS for 5 min each. The sample was then blocked in 3% BSA for 1 h after three washes in PBS/PVA. After blocking, all embryos were incubated with primary antibodies at 4 °C overnight. The primary antibodies used were mouse anti-*ZSCAN4* antibody (1:50; Cat # TA800425; Invitrogen), rabbit anti-*DNMT1* antibody (1:100; Cat # ab188453; Abcam, Cambridge, UK), rabbit anti-histone H3 (acetyl K27) antibody (1:100; Cat # ab177178; Abcam), rabbit anti-H3 (tri-methyl K9) antibody (1:100; Cat # ab8898; Abcam), rabbit anti-phospho-histone H2A.X antibody (1:100; Cat # 2577S; Cell Signaling Technology, MA, USA), rabbit anti-CASPASE-3 antibody (1:100; Cat # 9664S; Cell Signaling Technology), rabbit anti-p53 antibody (1:100; Cat # sc-6243; Santa Cruz Biotechnology, CA, USA), mouse anti-PML antibody (1:100; Cat # ab96051; Abcam), and rabbit anti-5mc antibody (1:200; Cat # 28692S; Cell Signaling Technology). After three washes in PBS/PVA, the embryos were incubated for 1 h with Alexa Fluor 546™ donkey anti-rabbit IgG (H + L) (1:200; Cat # A10040; Invitrogen) or Alexa Fluor 488™ donkey anti-mouse IgG (H + L) (1:200; Cat # A21202; Invitrogen) at room temperature. Hoechst 33342 was used to stain the chromosomes. Finally, all stained embryos were mounted on glass slides and examined under a laser scanning confocal microscope (LSM 710 META; Zeiss, Oberkochen, Germany). Images were obtained and analyzed using ImageJ software (version 1.53K, National Institutes of Health, Bethesda, MD, USA). 

### 3.6. Western Blot Analysis

Approximately 100 embryos in the four-cell stage from each group were collected after 48 h of culture. Western blot analysis was performed as previously reported [33]. Sodium dodecyl sulphate sample buffer (1×) was used to lyse the oocytes at 90 °C for 10 min. Proteins were separated using sodium dodecyl sulfate-polyacrylamide gel electrophoresis at 80 V for 90 min and transferred onto polyvinylidene fluoride membranes at 250 mA for 60 min. TBST containing 5% nonfat milk was used to block the membranes for 1 h. Next, the membranes were incubated with mouse anti-*ZSCAN4* antibody (1:50; Cat # TA800425; Invitrogen), rabbit anti-*DNMT1* antibody (1:100; Cat # ab188453; Abcam), rabbit anti-histone H3 (acetyl K27) antibody (1:100; Cat # ab177178; Abcam), rabbit anti-H3 (tri-methyl K9) antibody (1:100; Cat # ab8898; Abcam), rabbit anti-phospho-histone H2A.X antibody (1:100; Cat # 2577S; Cell Signaling Technology), and mouse anti-PML antibody (1:100; Cat # ab96051; Abcam) at 4 °C overnight. After washing five times with TBST (5 min each), the membranes were incubated with horseradish peroxidase-conjugated goat anti-mouse IgG or goat anti-rabbit IgG (1:20,000; Santa Cruz Biotechnology) for 1 h at room temperature. Finally, the membranes were exposed using a CCD camera and UVI Soft software (Alliance Q9, UVITEC Cambridge), and the results were analyzed using ImageJ software (National Institutes of Health).

### 3.7. Statistical Analysis

Four-cell embryos were used for immunofluorescence, Western blot, and q-PCR. All experiments were performed in three biological repeats. A total of 100 four-cell embryos were used for each Western blot and each q-PCR experiment. Each q-PCR was performed in three technical repeats. “n” represents the total number of embryos analyzed and mean ± standard error was calculated for the group results. All data were analyzed using an independent sample *t*-test or ANOVA with GraphPad Prism 5 software (GraphPad Software Inc., La Jolla, CA, USA). Statistical significance was set at *p* < 0.05.

## 4. Discussion

In this study, we aimed to investigate the function of *ZSCAN4* in the development of preimplantation porcine embryos. Our results showed that *ZSCAN4* is critical for transcriptional activation in the ZGA stage and maintaining telomere length and gene stability in porcine embryos. Knockdown of *ZSCAN4* leads to shortening of telomeres and DNA damage, which leads to the arrest of porcine embryonic development.

*ZSCAN4*, a ZGA gene, is specifically expressed in the two-cell stage of early mouse embryonic development [8]. The transcript level of *ZSCAN4* is significantly increased during the four- to eight-cell stages of human embryonic development [34,35]. In early-stage bovine embryos, the Mrna expression of *ZSCAN4* is maintained at high levels in the 8- and 16-cell stages [14]. In our study, *ZSCAN4* was highly expressed in the four- to eight-cell stages, and *ZSCAN4* expression significantly decreased in the morula and blastocyst stages. These results suggest that *ZSCAN4* may play an active role in the ZGA stage in porcine embryos.

To explore the potential functions of *ZSCAN4* during early porcine embryonic development, we knocked down *ZSCAN4* by injecting *ZSCAN4*-dsRNA. We found that *ZSCAN4* KD did not affect the formation of two-cell embryos; however, the formation of four-cell embryos, eight-cell embryos, and blastocysts was significantly reduced, and the quality of blastocysts after *ZSCAN4* KD was also severely affected. Our results indicate that *ZSCAN4* is essential for preimplantation porcine embryonic development. The transcript levels of repetitive sequences in constitutive heterochromatin are markedly increased in *ZSCAN4*-overexpressing mESCs [10]. Highly transcribed constitutive heterochromatin is required for early embryonic development [36,37]. Generally, gene activation means that the chromatin enters an open state; furthermore, active markers are enriched, and repressive markers are lost in gene regulatory regions [38,39]. In this study, we examined histone modifications involved in the regulation of transcriptionally active chromatin, namely histone H3 lysine 27 acetylation (H3K27ac). We found that the level of H3K27ac was decreased after *ZSCAN4* KD. *ZSCAN4*^+^ mESCs were found to have higher levels of H3K27ac than *ZSCAN4*^−^ cells [10]. In mESCs, histone methylation associated with heterochromatin, such as H3K9me3 and H3K20me3, is relocalized in larger and fewer clusters during the *ZSCAN4* burst [10]. We found that *ZSCAN4* KD embryos had higher levels of H3K9me3 than the control embryos. These results suggest that *ZSCAN4* regulates gene activation during ZGA in porcine embryos. DNA methylation is another important epigenetic modification that regulates gene expression. DNA methylation is usually associated with silencing of gene expression [40]. DNA methylation is also a key factor in the epigenetic regulation of mammalian embryonic development. During the ZGA stage in goat embryos, TET1 expression increases, whereas *DNMT1* expression and DNA methylation decrease [41]. In two-cell-like cells, MERVL/*ZSCAN4* network activation leads to transient genome-wide DNA demethylation. Transient DNA demethylation is driven by the loss of DNA methyltransferases, including the methyltransferase *DNMT1* and de novo methyltransferases DNMT3a/DNMT3b [42]. In addition, a previous study reported that human *ZSCAN4* recruits TET2 through its SCAN region to promote DNA demethylation [16]. This study showed that the KD of *ZSCAN4* upregulated *DNMT1* protein level and downregulated TET2 protein level, which then induced DNA methylation. These results suggest that the conserved functions of *ZSCAN4* in mediating DNA methyltransferases and TET2 expression.

Maintenance and elongation of telomere length during early embryonic development are critical for successful implantation [43]. Preimplantation embryos can undergo rapid telomere extension through telomere recombination or telomere sister chromatid exchange (T-SCE) [25]. In ES cells, overexpression of *ZSCAN4* extends telomeres and suppresses spontaneous sister chromatid exchange [9]. In this study, the KD of *ZSCAN4* resulted in telomere shortening in porcine four-cell embryos, which also suggests the function of *ZSCAN4* in maintaining telomere length in porcine embryos. The most important function of telomeres is to protect the ends of chromosomes from being recognized as DNA breaks and to prevent abnormal chromosomal recombination [44]. When telomeres are shortened, they initiate a DNA damage response and trigger the p53/pRb pathway, leading to irreversible cell cycle arrest [45,46]. *ZSCAN4* plays a crucial role in chromosomal integrity and genomic stability in early stage embryos and promotes the repair of DNA damage and correction of abnormal chromosomes [47]. Jiang et al. reported that *ZSCAN4* binds to the Yamanaka factor to reduce DNA-damage response, downregulates p53, and significantly improves the efficiency of induced pluripotent stem cell generation [48]. In porcine embryos, *ZSCAN4* knockdown exacerbated DNA damage and upregulated p53 expression. Additionally, *ZSCAN4* KD led to apoptosis. These observations suggest that the regulatory effect of *ZSCAN4* on DNA damage and gene stability may be based on the maintenance of telomere length. In addition, Srinivasan et al. proposed a new developmental regulation mechanism; that is, *ZSCAN4* binds to nucleosomes to protect the genome from DNA damage during embryogenesis, which is associated with high transcriptional burden and genomic stress [49]. This suggests that *ZSCAN4* regulates genome stability through multiple pathways.

DNA methylation is an essential component of telomere length regulation. We analyzed the potential mechanism of action of *ZSCAN4* in telomere length maintenance. We knocked down *DNMT1* to inhibit DNA methylation. We found that *DNMT1* KD effectively resulted in telomere shortening and embryonic developmental arrest caused by *ZSCAN4* KD. In mESCs, *ZSCAN4* recruits the UHRF1-*DNMT1* complex and promotes UHRF1-mediated ubiquitination of UHRF1 and *DNMT1* to induce the subsequent degradation of UHRF1 and *DNMT1* [26]. Our results demonstrate that *ZSCAN4* downregulates *DNMT1* expression to stabilize telomere length. The maintenance of telomere length during early embryonic development is regulated by the ALT pathway. A characteristic feature of this pathway is the assembly of ALT-associated PML nuclear bodies in telomeres. The assembly of ALT-positive cells induces telomere elongation [29]. In our study, the KD of *ZSCAN4* resulted in reduced expression of PML, whereas double KD of *DNMT1* and *ZSCAN4* rescued the defect in PML expression. These results suggest that *ZSCAN4* and *DNMT1* regulate telomere length through the ALT pathway in porcine parthenogenetic embryos (Figure 7).

Our study discovered the functions of *ZSCAN4* on telomere length and ZGA in porcine parthenogenetic embryos. It also made us think about the role of *ZSCAN4* in fertilized porcine embryos. In bovine embryos, knockdown of *ZSCAN4* leads to abnormal expression of ZGA gene [14]. In mouse two-cell embryos, activation of *ZSCAN4* is critical for telomere elongation [18]. These results suggested that *ZSCAN4* may have similar functions in porcine embryos through in vitro fertilization. Our results may advance the understanding of the mechanisms by which *ZSCAN4* regulates the in vitro developmental potential of embryos. In addition, the underlying mechanism of how *ZSCAN4* affects histone modifications (histone acetylation and methylation) is unclear. In the future, exploring the relationship between *ZSCAN4* and histone modifying enzymes may help us understand the regulatory mechanism of *ZSCAN4* in porcine embryos.

## 5. Conclusions

In summary, this study reveals the functions of *ZSCAN4* in porcine parthenogenetic embryos. Importantly, *ZSCAN4* is involved in the regulation of transcriptional activity during the ZGA stage and stabilizes telomere length by downregulating *DNMT1*. These findings may provide new insights into the mechanism of telomere lengthening and epigenetic modification in porcine parthenogenetic embryos.

## Figures and Tables

**Figure 1 ijms-24-12121-f001:**
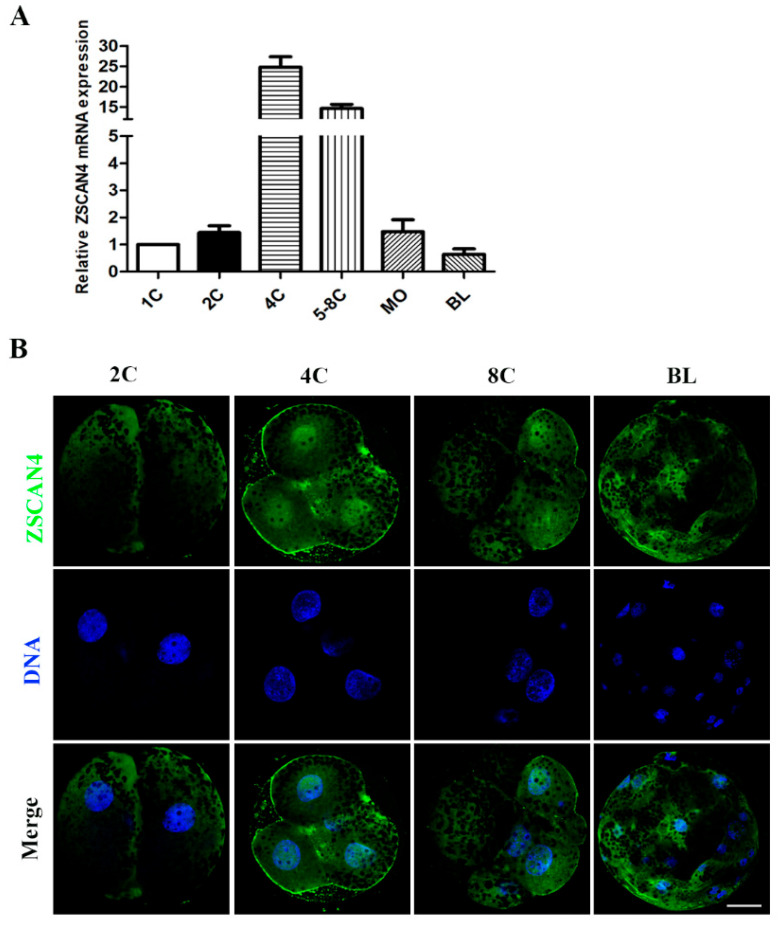
Expression and localization of *ZSCAN4* in porcine embryos. (**A**) qRT-PCR assay of the expression of *ZSCAN4* in different stages (one-, two-, four-, and five- to eight-cell embryos, morula, blastocysts) during porcine embryonic development; *18S* was selected as the reference gene. (**B**) Embryos in the two-, four-, and eight-cell and blastocyst stages were immunolabeled with anti-*ZSCAN4* (green); Hoechst 33342 was used to label DNA (blue). Bar = 20 μm.

**Figure 2 ijms-24-12121-f002:**
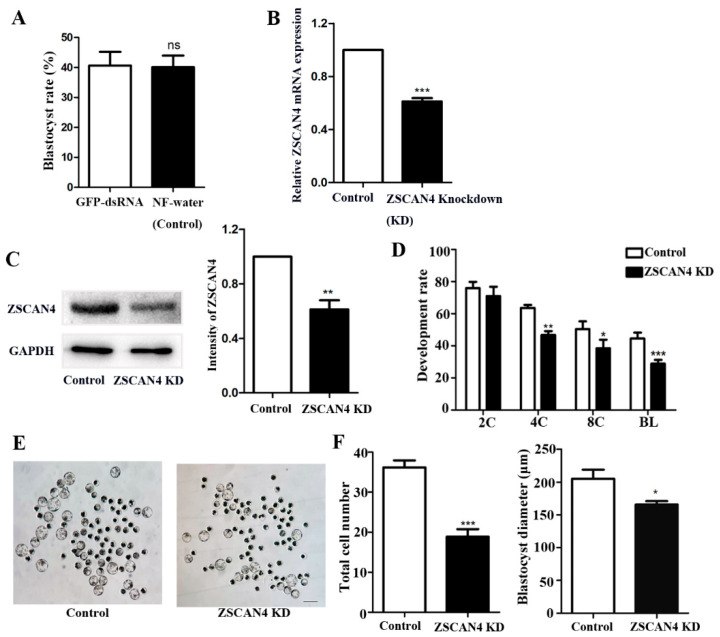
Effects of *ZSCAN4* KD on porcine embryonic development. (**A**) Blastocyst rate of embryos in the GFP-dsRNA injection group (*n* = 252) and the NF-water injection group (n = 251). (**B**) qRT-PCR assay was conducted to confirm *ZSCAN4* KD in the four-cell stage. (**C**) Western blotting was conducted to confirm *ZSCAN4* KD in the four-cell stage. (**D**) Development rate of embryos in the two-cell, four-cell, eight-cell, and blastocyst stages in the control group and *ZSCAN4* group. (**E**) The morphology of embryos after 7 days of in vitro culture in the control group (*n* = 9) and *ZSCAN4* KD group (*n* = 9). Bar = 100 μm. (**F**) Total cell number and diameter of blastocyst in the control group (*n* = 14) and the *ZSCAN4* KD group (*n* = 19). Results with * *p* < 0.05, ** *p* < 0.01 and *** *p* < 0.001 were considered significant.

**Figure 3 ijms-24-12121-f003:**
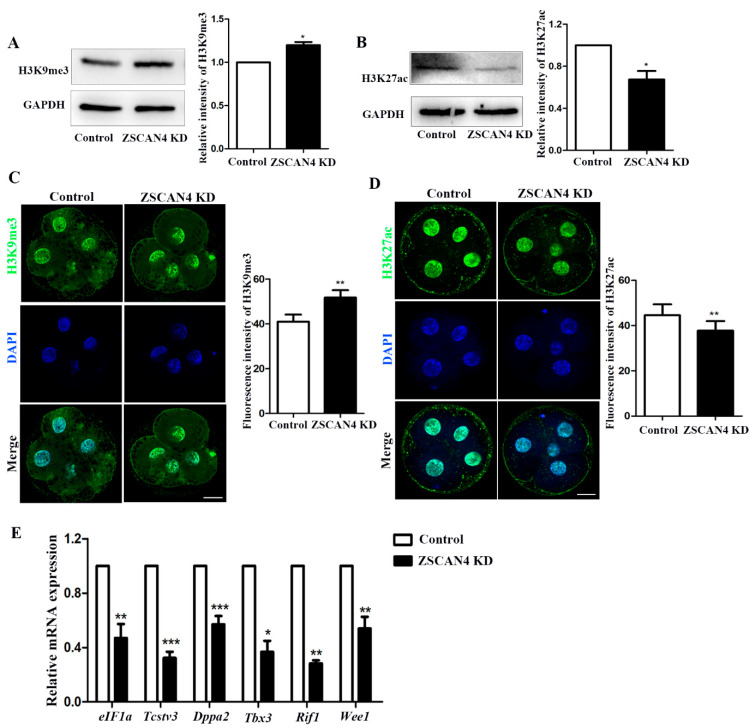
Effects of *ZSCAN4* KD on histone modification and ZGA. (**A**) Western blotting was used to determine H3K9me3 level in the four-cell stage. (**B**) Western blotting was used to determine H3K27ac level in the four-cell stage. (**C**) Embryos in the four-cell stage were immunolabeled with anti-H3K9me3 (green); Hoechst 33342 was used to label DNA (blue). Bar = 20 μm. (**D**) Staining images of H3K27ac in control four-cell embryos (*n* = 71) and *ZSCAN4* KD four-cell embryos (*n* = 67). (**E**) qRT-PCR assay was used to determine the mRNA level of *eIF1a*, *Tbx3*, *Tcstv3*, *Rif1*, *Wee1*, and *Dppa2* in the four-cell stage. Green, H3K27ac; blue, DNA. Bar = 20 μm. Results with * *p* < 0.05, ** *p* < 0.01 and *** *p* < 0.001 were considered significant.

**Figure 4 ijms-24-12121-f004:**
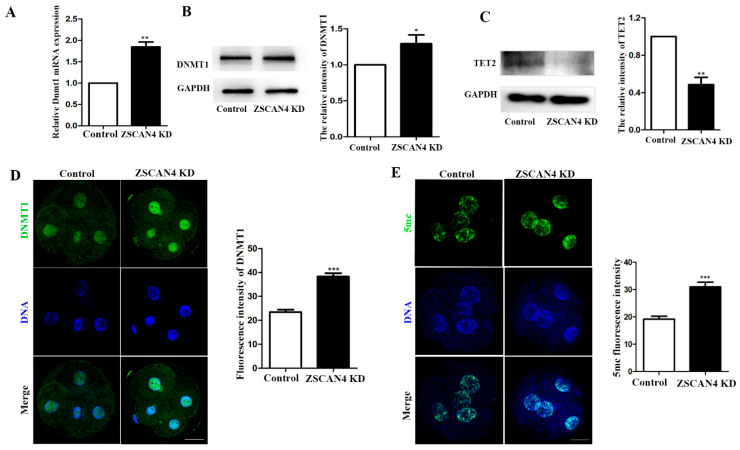
*ZSCAN4* KD induced global DNA methylation. (**A**) qRT-PCR assay was used to determine the *DNMT1* mRNA level in the four-cell stage. (**B**) Western blotting was conducted to determine *DNMT1* protein level in the four-cell stage. (**C**) Western blotting was used to determine TET2 protein level in the four-cell stage. (**D**) Staining images of *DNMT1* in control four-cell embryos (*n* = 40) and *ZSCAN4* KD four-cell embryos (*n* = 32). Green, *DNMT1*; blue, DNA. Bar = 20 μm. (**E**) Embryos in the four-cell stage were immunolabeled with anti-5mc (green); Hoechst 33342 was used to label DNA (blue) (Control: *n* = 45, *ZSCAN4* KD: *n* = 43). Bar = 20 μm. Results with * *p* < 0.05, ** *p* < 0.01 and *** *p* < 0.001 were considered significant.

**Figure 5 ijms-24-12121-f005:**
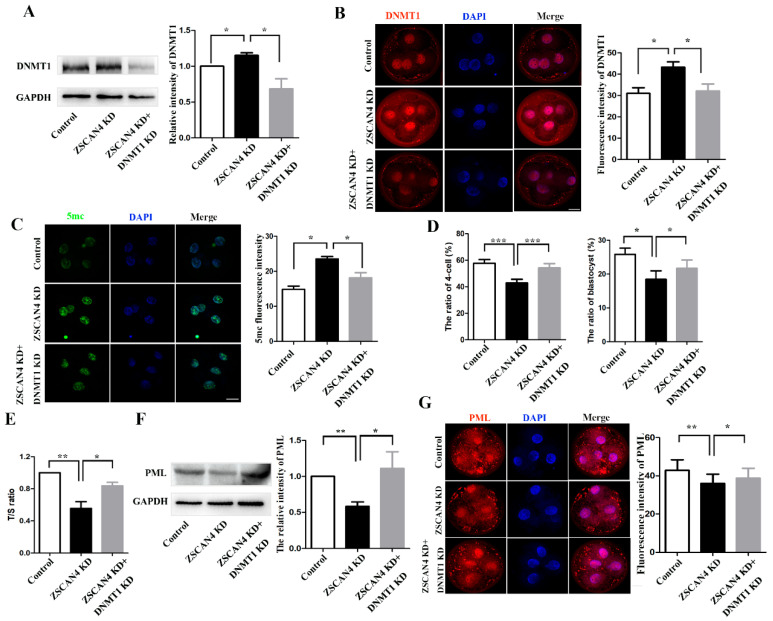
*ZSCAN4* regulated *DNMT1* expression to stabilize telomere length. (**A**) Band intensity analysis of *DNMT1* in the four-cell stage after *ZSCAN4* KD and *DNMT1* KD. (**B**) Images and fluorescence intensity of *DNMT1* in control embryo (*n* = 130), *ZSCAN4* KD embryos (*n* = 133), and *ZSCAN4* KD + *DNMT1* KD embryos (*n* = 129). Red: *DNMT1*; blue: DNA. Bar = 20 μm. (**C**) Images and fluorescence intensity of 5mc in control embryo (*n* = 75), *ZSCAN4* KD embryos (*n* = 84), and *ZSCAN4* KD + *DNMT1* KD embryos (*n* = 61). Green: 5mc; blue: DNA. Bar = 20 μm. (**D**) Four-cell and blastocyst ratio after *ZSCAN4* KD and *DNMT1* KD. (**E**) qPCR assay for relative telomere length in the four-cell stage. Pig *36B4* single copy gene was used as the control gene. (**F**) Band intensity of PML in the four-cell stage after *ZSCAN4* KD and *DNMT1* KD. (**G**) Images and fluorescence intensity of PML in the control group (*n* = 138), *ZSCAN4* KD group (*n* = 104), and *ZSCAN4* KD + *DNMT1* KD group (*n* = 104). Red: PML; blue: DNA. Bar = 20 μm. Results with * *p* < 0.05, ** *p* < 0.01 and *** *p* < 0.001 were considered significant.

**Figure 6 ijms-24-12121-f006:**
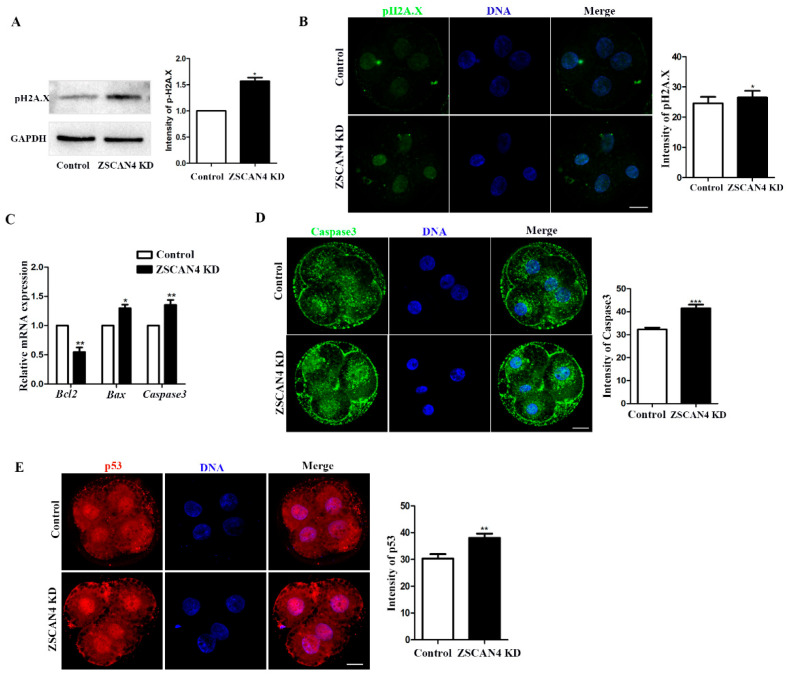
*ZSCAN4* knockdown induced DNA damage and apoptosis in porcine embryos. (**A**) Western blotting was used to determine the level of pH2A.X in the four-cell stage. (**B**) Images and fluorescence intensity of pH2A.X in control embryos (*n* = 126) and *ZSCAN4* KD embryos (*n* = 101). Green, pH2A.X; blue, DNA. Bar = 20 μm. (**C**) qRT-PCR assay was used to determine the mRNA level of *Bcl2*, *Bax*, and *Caspase3* in the four-cell stage. (**D**) Images and fluorescence intensity of *Caspase3* in control embryos (*n* = 58) and *ZSCAN4* KD embryos (*n* = 51). Green, *Caspase3*; blue, DNA. Bar = 20 μm. (**E**) Staining images of p53 in control embryos (*n* = 73) and *ZSCAN4* KD embryos (*n* = 58). Red, p53; blue, DNA. Bar = 20 μm. Results with * *p* < 0.05, ** *p* <0.01 and *** *p* < 0.001 were considered significant.

**Figure 7 ijms-24-12121-f007:**
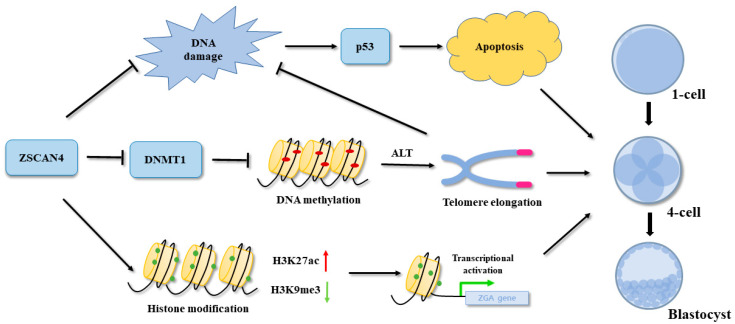
Schematic representation depicting the functions of *ZSCAN4* in porcine embryos. *ZSCAN4* promoted transcription by regulating histone acetylation (H3K27ac) and methylation (H3K9me3) in porcine ZGA. Knockdown of *ZSCAN4* induced DNA damage and apoptosis. Moreover, *ZSCAN4* inhibited *DNMT1* expression, causing DNA demethylation. Low levels of DNA methylation can promote telomere elongation through the ALT pathway.

**Table 1 ijms-24-12121-t001:** Primer sequences.

Gene	Forward Primer	Reverse Primer
ds-*ZSCAN4*	5′-GCCCTCTTTTCTGAGAATATGCC-3′	5′-CTGATGGACTTTCAACCGAGA-3′
ds-*DNMT1*	5′-CAAACTACCAGGCAGACCAC-3′	5′-CACTACTGCCGTTTTGGTTCG-3′
*ZSCAN4*	5′-CTTGTTTGGTCCTCGAACAGT-3′	5′-TTCATGCCATCGTCTGTCAGGT-3′
*DNMT1*	5′-CTCCCTACAGAAGAACCGGAA-3′	5′-CCTCGTGCTTCTGTCTAGCTC-3′
*Telomere*	5′-GGT TTT TGA GGG TGA GGG TGA GGG TGA GGG TGA GGG T-3′	5′-TCC CGA CTA TCC CTA TCC CTA TCC CTA TCC CTA TCC CTA-3′
*36B4*	5′-TGAAGTGCTTGACATCACCGAGGA-3′	5′-CTGCAGACATACGCTGGCAACATT-3′
*Caspase3*	5′-TCTAACTGGCAAACCCAAACTT-3′	5′-AGTCCCACTGTCCGTCTCAAT-3′
*BAX*	5′-GAATGGGGGGAGAGACACCT-3′	5′-CCGCCACTCGGAAAAAGA-3′
*BCL2*	5′-GAACTGGGGGAGGATTGTGG-3′	5′-CATCCCAGCCTCCGTTATCC-3′
*18s*	5′-CGCGGTTCTATTTTGTTGGT’-3′	5′-AGTCGGCATCGTTTATGGTC-3′
*TCSTV3*	5′-AGAAAGGGCTGGAACTTGTGACCT-3′	5′-AAAGCTCTTTGAAGCCATGCCCAG-3′
*WEE1*	5′-ACCTCGGATTCCACAAGTGCTTT-3′	5′-ATGCTTTACCAGTGCCATTGCT-3′
*RIF1*	5′-TTGACATCATTTTACCGCAGA-3′	5′-GGAATCTCTTCTAGTCCACGA-3′
*TBX3*	5′-CTGACCCCGAAATGCCGAAG-3′	5′-ACTATAATTCCCCTGCCACGTT-3′
*EIF1A*	5′-GGTGTTCAAAGAAGATGGGCAAGAG-3′	5′-TTTCCCTCTGATGTGACATAACCTC-3′
*DPPA2*	5′-TACAGAAGGTTGGGTTCGCC-3′	5′-GGTCTGGGGATGGGAAAGTG-3′

## Data Availability

The data generated in this study have been included in this article.

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
