# Peer review of "ZSCAN4 Regulates Zygotic Genome Activation and Telomere Elongation in Porcine Parthenogenetic Embryos"

_ijms, 2023, doi:10.3390/ijms241512121_

Round 1
Reviewer 1 Report
1. L19, please define 5mC.
2. L53, there is no need to have “HNSCC” abbreviation since it only appears once in the manuscript.
3. L80, please list the city, state, country of Sigma-Aldrich.
4. L87, MEGAscript is an in vitro (not an in vivo) transcription kit.
5. L109, please indicate the brand name of the four-well plates used. “370 mL/well” >> 370 µL/well.
6. L117-118, please describe the injection procedure and the concentration of dsRNA of the injection solution.
7. L119, PZM-5 is an in vitro culture medium.
8. L131-136, qPCR, rather than qRT-PCR, is used to perform telomere measurement. Please make corrections throughout the manuscript.
9. L142, the embryos were fixed in 70% ethanol. Why PBS was aspirated here?
10. L292, “to determined” >> to determine.
11. L358-359, apoptosis is the process of programmed cell death rather than cell inactivation.
12. L402, only preimplantation embryos (zygotes to blastocysts) were examined, therefore, “implanted” >> preimplantation.
13. L419-420, “In two-cell like mESCs”?
14. L454, “rescued” >> resulted in?
Please carefully read your manuscript before resubmission.
Reviewer 2 Report
In the presented manuscript, the Authors investigated the role of Zinc finger and SCAN domain-containing 4 (ZSCAN4) in early development of porcine embryos. The study is very complex, gene and protein expression of multiple factors related to genome activation, telomeres, DNA methylation and apoptosis were investigated. Some of these aspects have been studied in mice and bovine, but not in pig and not in such extensive manner. Therefore the manuscript provides novel, comprehensive data.
The manuscript is well written, but the structure is somewhat disturbed (see minor notes). The introduction part provides sufficient background. The aim is clear and the results are well presented, with multiple figures and photographs, which enrich the manuscript. The data are discussed properly and the conclusions are supported by the results.
However, I see one major methodological issue: the study was performed on parthenogenetically derived embryos. Multiple studies in many species showed aberrant gene expression and epigenetic impairment, resulting in abnormal development in parthenogenetic embryos. As the study was focused on the ZSCAN4, a regulatory factor, this developmental differences could have an impact. The Authors should explain, why they chose such model instead of classic in vitro fertilization. Also, it should be discussed if the data can be valid for ‘normal embryos’ as well. In the title, abstract and through the manuscript it should be pinpoint that data were obtained on such material (e.g. instead ‘preimplantation embryos’ – ‘parthenogenetic embryos’).
Other, small issues in M&M section can be improved as well.
I believe that after answering this major issue, this manuscript will be suitable for publication in International Journal of Molecular Sciences.
Minor remarks:
Lines 75-79: results should not be included in the aim paragraph.
Line 103-108: 80 COCs per plate or per 500 µl well? Now it sounds like in the whole plate there was 500 µl of medium with 80 COCs.
Line 121: PA embryos were cultured for how many days?
Line 130: a reference gene should be provided here, not only in the figures.
Line 180: ‘more than three times’ is very imprecise. Study design paragraph should be added, with information how many biological replicates (how many procedures of parthenogenetic activation and embryoculture and till which stage) were performed for each step and how many embryos were used. ‘N’ number should be included also in the figure description. Number of technical replicates for qPCR and Western Blotting should be provided as well.
Line 187-188 and further: each result paragraph starts with short description of a particular step, sometimes with references, which is not a classic manuscript structure. For more traditional style, description should be moved to the study design paragraph, although I leave it to the Editor’s decision, if a present form suit the Journal criteria.
Figure 1A and further: instead of ‘RNA level’ should be relative RNA expression (as in Fig.3)
Figure 2: figures should be understandable without reading the main text, therefore abbreviations (like ‘KD’) should be avoided, or at least for the first time the full word should appear.
Figure 5: they are too small and hard to read – should be similar size as the previous figures
Line 402: ‘implanted’?
Line 442: an abbreviation DDR is not explained
Line 443: an abbreviation iPS is not explained
Reviewer 3 Report
In this manuscript, the authors investigated the function of ZSCAN4 in the development of preimplantation porcine embryos via dsRNA injection. The results showed that ZSCAN4 plays an important role in the ZGA stage and maintaining telomere length and gene stability. Here are some concerned need to be clarified.
Majors:
1. For dsRNA injection, the random dsRNA should be used as control instead of nucleus-free water.
2. I suggest the author doing rescue experiments via ZSCAN4 cRNA injection, which would confirm the direct causality of the ZSCAN4 and phenotype.
3. For Figure 3A, the data is not clear and convincing enough since the control band looks like uneven distributed for the blot imaging step. Could the author provide some other images?
Minors:
1. Table 1, it would be clearer if the author could add 5’-3’ direction for the primers.
2. Line 232, ‘in vitro’ should be shown as italic.
3. The author should give a deeper discussion about the limitation for this research.
Round 2
Reviewer 2 Report
All my comments have been addressed satisfactory and the manuscript has been corrected accordingly. I have no more notes - the manuscript can be published in the International Journal of Molecular Sciences.
Author Response
Thank you very much for your comments and suggestions.
Reviewer 3 Report
For major concern 1, if the author injected GFP-dsRNA as control, the correpsonding data should be shown in the figure. Other points that I raised have been addressed properly. I agree to publish this paper after minor revision.
